# Blocks: Creating Rich Tables with Drag-and-Drop Interaction

Category: Research

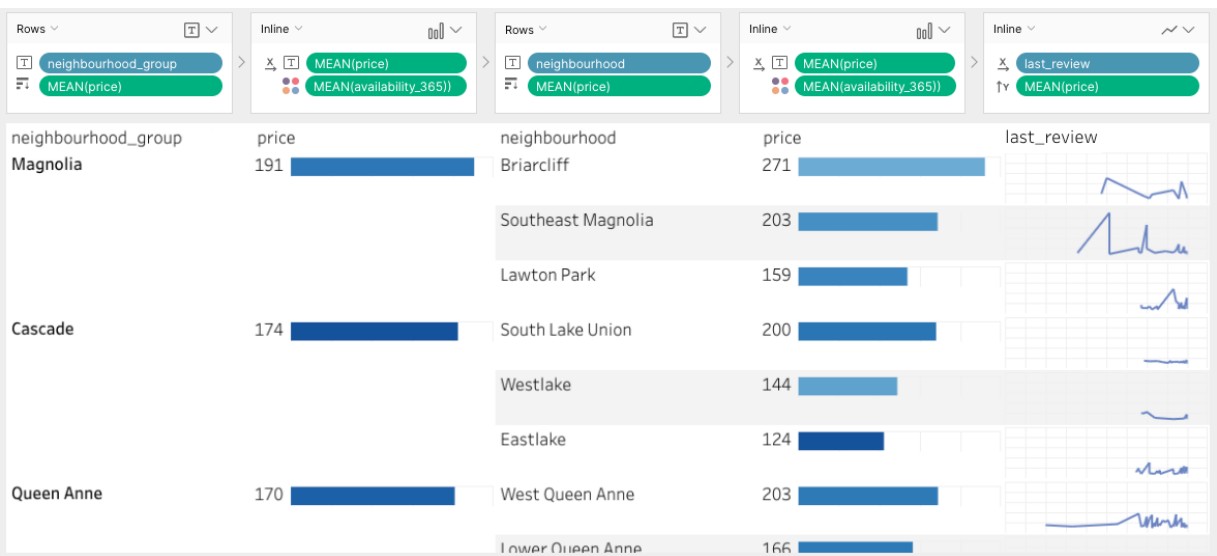

Figure 1: A rich table showing data about Airbnb listings in Seattle, created with Blocks. The table shows a variety of mark types and measures at several levels of detail combined into a single visualization. Each column of the table is defined by a Block with its own set of encoding and field mappings. The columns from left to right show rows for each neighborhood group, sorted by average listing price, a labeled bar chart showing average price, colored by availability, rows for each neighbourhood within each neighborhood group, the same labeled bar chart, but showing average price for each neighborhood, and a sparkline showing average price over time.

## ABSTRACT

We present *Blocks*, a formalism that enables the building of visualizations by specifying layout, data relationships, and level of detail (LOD) for specific portions of the visualization. Users can create and manipulate Blocks on a canvas interface through drag-and-drop interaction, controlling the LOD of the data attributes for tabular style visualizations. We conducted a user study to compare how 24 participants employ Blocks and Tableau in their analytical workflows to complete a target visualization task. We also ran a subsequent longitudinal diary study with eight participants to better understand both the usability and utility of Blocks in their own analytical inquiries. Findings from the study suggest that Blocks is a useful mechanism for creating visualizations with embedded microcharts, conditional formatting, and custom layouts. We finally describe how the Blocks formalism can be extended to support additional composite visualizations and Sankey charts, along with future implications for designing visual analysis interfaces that can handle creating more complex charts through drag-and-drop interaction.

**Keywords:** Formalism, level of detail, nesting, layout, conditional formatting, rich tables, drag-and-drop interaction.

## 1 INTRODUCTION

Visual analysis tools [15, 23] help support the user in data exploration and iterative view refinement. Some of these tools are more expressive, giving expert users more control, while others are easier to learn and faster to create visualizations. These tools are often driven by underlying grammars of graphics [27, 43] that provide various formalisms to concisely describe the components of a visualization. High level formalisms such as VizQL [40] and ggplot2 [42] are set up to support partial specifications of the visualization and hence provide the convenience of concise representations. Reasonable defaults are subsequently applied to infer missing information to generate a valid graphic. The downside of these concise representations is that the support for expressiveness for visualization generation in these tools is either limited or difficult for a user to learn how to do.

Drag-and-drop is one paradigm for addressing the limitations of expressivity by supporting task expression through user interaction where the visibility of the object of interest replaces complex language syntax. VizQL is one such formalism that supports the expression of chart creation through direct manipulation in Tableau [23]. While the language enables users to create charts through its underlying compositional algebra, there is still tight coupling between the query, the visualization structure, and layout. As a result, users often spend significant time in generating complex visualizations when they have a specific structure and layout in mind. The other paradigm for promoting expressiveness for chart creation is through the use of declarative specification grammars [29, 38, 39] that can programmatically express the developer's intentions.

Despite the prevalence of these tools, creating expressive data visualizations still remains a challenging task. Beyond having a good insight about how the data can be best visualized, users need to have sufficient knowledge to generate these visualizations. So, how can we support users in their analytical workflows by enabling a greater degree of flexibility and control over nesting relationships, layout, and encodings, yet providing the intuitiveness of a user interface? In this paper, we address this dichotomy between expressibility and ease of use for the user by extending VizQL to provide greater flexibility in creating expressive charts through direct manipulation.

## 1.1 Contributions

Specifically, our contributions are as follows:

- We introduce *Blocks*, a formalism that builds upon VizQL by supporting the nested relationships between attributes in a visualization using a drag-and-drop interaction. Every component of the visualization is an analytical entity to which different nesting and encoding properties can be applied.

- We implement a *Blocks System* that provides a user increased flexibility with layout and formatting options through the direct manipulation of Block objects in the interface.

- We evaluated Blocks with 24 participants when performing tasks involving the creation of rich tables using both Tableau and Blocks. Eight of these users recorded their explorations using Blocks in their own workflows for an additional two-week diary study. Findings from the studies indicate that Blocks is a promising paradigm for the creation of complex charts. We identify research directions to pursue to better support users' mental models when using the system.

Figure 1 shows how a user can create a rich table using Blocks with a Seattle Airbnb dataset. The assembly of Blocks in the interface results in columns with different mark types such as bar charts and sparklines. The query for each Block inherits the dimensions from the parent Blocks. The first price column inherits the field `neighbourhood_group` as its dimension, computing price for each neighbourhood group. The second price column inherits both `neighbourhood_group` and the field `neighbourhood` showing a more granular level of `price` per `neighbourhood`.

## 2 RELATED WORK

Visual analysis techniques can be broadly classified into two main categories: (1) declarative specification grammars that provide high-level language abstractions and (2) visual analysis interfaces that facilitate chart generation through interaction modalities.

### 2.1 Declarative specification grammars

Declarative visualization languages address the problem of expressiveness by allowing developers to concisely express how they would like to render a visualization. Vega [39] and Vega-Lite [38] support the authoring of interactivity in the visualizations. While these specification languages provide a great degree of flexibility in how charts can be programmatically generated, they provide limited support for displaying different levels of granularity within a field in a visualization. Further, they require programming experience, making it challenging for non-developers to quickly develop advanced charts in their flow of analysis. Viser [41] addresses this gap by automatically synthesizing visualization scripts from simple visual sketches provided by the user. Specifically, given an input data set and a visual sketch that demonstrates how to visualize a very small subset of this data, their technique automatically generates a program that can be used to visualize the entire data set. Ivy [35] proposes parameterized declarative templates, an abstraction mechanism over JSON-based visualization grammars. A related effort by Harper and Agrawala [32] converts D3 charts into reusable Vega-Lite templates for a limited subset of D3 charts. While our work is similar to that of declarative grammars and template specifications in the sense of abstracting low-level implementation details from the user, we focus on supporting non-developer analysts in creating expressive charts through drag-and-drop interaction. We specifically extend the formalism of VizQL for supporting nested queries, layout, and encoding flexibility through drag-and-drop interaction in the Blocks interface.

## 2.2 Visual analysis interfaces

Visual analysis tools over the years have developed ways to help novice users in getting started in a UI context. The basic form of these tools for chart generation include chart pickers that are prevalent in various visual analysis systems [26]. Commercial visual analysis tools such as Tableau and PowerBI, along with systems like Charticulator [36] are built on a visualization framework that enables users to map fields to visual attributes using drag-and-drop interaction. As more analytical capabilities are enabled in these tools, there is a disconnect from the underlying abstraction, leading to calculation editors and dialog menus that add both complexity and friction to the analytical workflow.

Prior work has explored combinations of interaction modalities for creating visualizations. Liger [37] combines shelf-based chart specification and visualization by demonstration. Hanpuku [28], Data-Driven Guides [33], and Data Illustrator [34] combine visual editor-style manipulation with chart specification. However, none of these systems specifically focus on a visually expressive way of handling nested relationships during chart generation; a common and important aspect of analytical workflows. Our work specifically addresses this gap and focuses on supporting analysts in a visual analysis interface for creating more expressive charts with nestings by using drag-and-drop as an interaction paradigm.

Domino [31] is a system where users can arrange and manipulate subsets, visualize data, and explicitly represent the relationships between these subsets. Our work is similar in concept wherein direct manipulation is employed in visually building relationships in charts, but there are differences. Domino has limited nesting and inheritance capabilities as it does not define parent-child relationships between blocks to support dependent relationships (e.g., a column depending on rows). The expressiveness of complex visualizations such as rich tables with repeated cells containing sparklines, text, and shapes, is limited.

## 3 TABLEAU USER EXPERIENCE

The core user experience of Tableau is placing *Pills* (data fields) onto *Shelves* (specific drop targets in the interface). This controls both the data used and the structure, along with the layout of the final visualization. Fields without an aggregation are called *Dimensions*. *Measures* are fields that are aggregated within groups defined by the set of all dimensions, i.e., the Level of Detail (LOD).

The key shelves are the *Rows Shelf*, the *Columns Shelf*, and the visual encoding shelves that are grouped into the *Marks Card*. Fields on the Rows and Columns Shelves define "headers" if discrete or "axes" if continuous. The Marks Card specifies a mark type and visual encodings such as size, shape, and color. If there is more than one Marks Card, the group of visualizations defined by the Marks Cards, forms the innermost part of the chart structure, repeated across the grid defined by the Rows and Columns Shelves.

The Blocks system attempts to address three limitations inherent to the Tableau experience:

- **The separation between "headers" and "marks" concepts.** The headers define the layout of the visualization and cannot be visually encoded. Only fields on the Marks Card participate in creating marks, but the marks must be arranged within the grid formed by the headers. For example, it is not possible to have a hierarchical table where the top level of the hierarchy is denoted by a symbol rather than text.

- **The Rows and Columns Shelves are global**. As per their names, a field on the Rows Shelf defines a horizontal band, and a field on the Columns Shelf a vertical band, across the entire visualization. For example, it is not possible to place a y-axis next to a simple text value, as one does for sparklines.

- **Queries are always defined using both the Rows and Columns Shelves, along with the Marks Card**. For example,

it is not possible to get the value of a measure at an LOD of only dimensions from the Rows Shelf, without those on the Columns Shelf.

Users have found ways to work around these limitations to build complex visualizations such as rich tables with sparklines or visualizations with encodings at different LOD for example. These methods include composing multiple visualizations on a dashboard so they appear as one [2]; writing complex calculations to control layout or formatting of elements [3–5, 7, 11–13]; creating axes with only a single value [1, 20], among others. Tableau introduced LOD expressions to help answer questions involving multiple levels of granularity in a single visualization [14]. The concept of LOD expressions is outside of the core UI paradigm of direct manipulation in Tableau. Rather, users need to define LOD calculated fields via a calculation editor and understand the syntax structure of Tableau formulae.

## 4 DESIGN GOALS

To better understand the limitations of Tableau for creating more expressive visualizations, we interviewed 19 customers, analyzed 7 internal dashboards, and reviewed 10 discussions on the Tableau Community Forums [18] that used various workarounds to accomplish their analytical needs. Each customer interview had one facilitator and one notetaker. The customers we interviewed consisted of medium- or large-sized companies that employ Tableau in their work. The interviews consisted of an hour-long discussion where we probed these customers to better understand their use cases. We conducted a thematic analysis through open-coding of interview notes and the Tableau workbooks the customers created and maintained. Finally, we reviewed the top ideas in the Tableau Community Forums to locate needs for more expressive visualizations. These ideas included extensive discussions among customers, which helped us better understand the use cases as well as ways customers work around limitations today. We reviewed our findings, summarized what we learned, and identified common patterns from our research. This analysis is codified into the following design goals:

**DG1. Support drag-and-drop interaction**
Tableau employs a drag-and-drop interface to support visual analysis exploration. We learned through discussions with an internal analyst how important table visualizations were for her initial exploration of her data. Her first analytic step was to view her data in a table at multiple LOD and confirming that the numbers matched her expectations based on domain knowledge. We also noticed that many customers used tables to check the accuracy of their calculations throughout their analysis. These discussions indicated that tables are not just an end goal of analysis, but play a key part of the exploratory drag-and-drop process. Our goal is to maintain the ease of use provided by the drag-and-drop interface and data-driven flow when creating visualizations.

**DG2. Better control over visualization components and layout**
Tableau employs defaults to help users manage the large space of possibilities that a compositional language creates [40]. When users have specific ideas of what they want to create, their workflows often conflict with the system defaults. A customer at a large apparel company described the challenges they ran into when replicating an existing report in Tableau. In order to match all of the desired formatting and layout, they had to delicately align multiple sheets together on a single dashboard. Not only did the customer find this frustrating to maintain, but they often ran into issues with alignment and responsive layout. Our goal is to support users with increased layout flexibility as they generate charts for their analytical needs.

**DG3. Aggregate and encode at any LOD in a visualization**
As users strive to build richer visualizations, the need arises for more control over showing information at multiple LOD. While Tableau supports calculations to control the LOD a measure aggregates to, creating these calculations does not provide the ability to visually encode at any LOD and takes users out of their analytic workflow. For example, one customer at a large technology company had a table visualization that listed projects and the teams who worked on each of the projects. Some of the measures needed to show information at the project level (such as total cost), while others measures were at the team level (amount of effort required per team). Building this visualization in Tableau required the customer to write many LOD calculations. Our goal is to provide the ability to use visual encodings and a drag-and-drop experience to evaluate measures at any LOD from any component of the visualization.

## 5 THE BLOCKS FORMALISM

The Blocks formalism uses an arbitrary number of connected local expressions (i.e., *Blocks*) instead of global Rows and Columns expressions. Each Block represents a single query of a data source at a single LOD, resulting in a component of the final visualization. Parent-child relationships between the Blocks form a directed acyclic graph (DAG).

A `block-name` is a unique identifier for the Block. The valid values of `field-name` and `aggregation` depend on the fields in the data source and the aggregation functions supported by that data source for each field. Any `field-instance` with an `aggregation` is used as a measure; all others are used as dimensions.

The *local LOD* of the Block is the set of all dimensions used by any `encoding` within the Block. The *full LOD* of the Block is the union of its local LOD and the local LOD of all of its ancestors. All of the measures used by the Block are evaluated at the full LOD of the Block. In addition to defining the LOD, the `encodings` map the query results to visual and spatial encodings. Except for ⬇ (sort ascending), ⬆ (sort descending), and ∴ (data details), each `encoding-type` must occur at most once within each Block. The sort encodings control the order of the query result and ultimately the rendering order; their priority is determined by the order that they appear. By providing a means to encode ⤬ (x-axis) and ↑ʏ (y-axis) at the visualization component level instead of as part of a global table expression as in Tableau, Blocks addresses **DG3** with respect to sparklines and other micro charts within a table visualization.

```
block           := (block-name, layout-type,
                    mark-type, encoding, children)
children        := {(child-group)}
child-group     := {block-name}
layout-type     := "rows" | "columns" | "inline"
mark-type       := "text" | "shape" | "circle"
                    | "line" | "bar"
encoding        := ({encoding-type},
                    field-instance)
encoding-type   := ⬤ "color" | ⬡ "size"
                    | ⬚ "shape"
                    | ⊡ "text" | ⤬ "x-axis"
                    | ↑ʏ "y-axis" | ⬇ "sort-asc"
                    | ⬆ "sort-desc" | ∴ "detail"
field-instance  := ([aggregation], field-name)
```

Each Block renders one mark of its `mark-type` per tuple in its query result. The `layout-type` determines how each of the Block's rendered marks are laid out in space. A Block with the layout type of `rows` creates a row for each value in its domain, with each row containing a single mark. A common example is a Block with a `rows` layout type and `text` mark type will generate a row displaying a text string for each value in the Block's domain. A Block with the layout type of `columns` creates a column for each value, each column

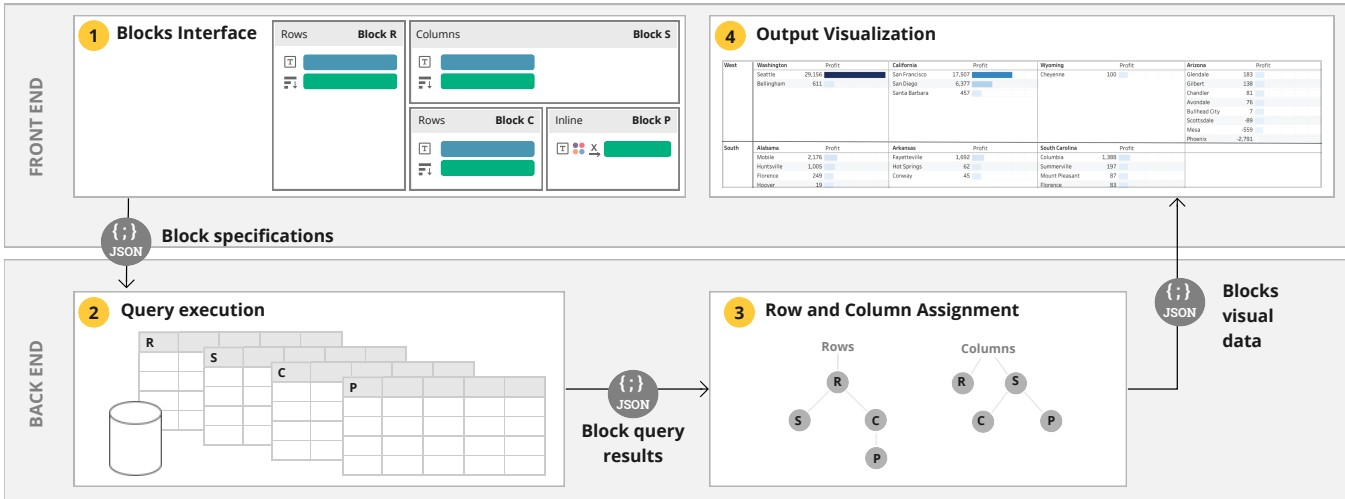

Figure 2: Blocks system overview. Users create *Block GUI Cards* that can define multiple field encodings at a single LOD. The *Block GUI card* is translated into a *Block specification*. This specification consists of some number of dimensions, some number of measures aggregated to the LOD of the cross product of the dimensions, a layout, the visual encodings, a mark type, some number of filters, and a sort order. From this *Block specification*, a *Block query* is issued to the source data source. The output of a *Block query* is a *Block result set* which returns the tuples and corresponding encoding results. This is finally rendered as an *output visualization*.

containing a single mark per column. To facilitate the creation of scatter plots, line graphs, area charts, and maps, a Block with the layout type of `inline` renders all of its marks in a single shared space.

Child Blocks are laid out in relation to their parents' positioning. A `child-group` is a set of children that share the same row (for a `rows` parent) or column (for a `columns` parent). E.g., in Figure 4d, the children of Block *R* are ((Block *B*, Block *C*), Block *G*); *B* and *C* are on the same row and so form a `child-group`. To insure the layout can be calculated, the DAG must simplify to a single tree when considering only the children of `rows` Blocks or only the children of `columns` Blocks. This layout system enables Blocks to address **DG2** by defining labels, axes, and marks all using the single Block concept. Figure 4a shows how Blocks can be expressed with the formalism.

## 6 THE BLOCKS SYSTEM

The Blocks system provides an interface for creating Blocks and to view the resulting visualizations. Figure 2 illustrates the architecture. The Blocks Interface (1) and Output Visualization (4) are React-based [17] TypeScript [25] modules that run in a web browser. The interface communicates over HTTPS with a Python back-end that implements the Query Execution (2) and Rows and Column Assignment (3) processes. The system has the flexibility of using either of two query execution systems – a simple one built on Pandas [16] and local text files, or a connection to a Tableau Server Data Source [22], which provides access to Tableau's rich data model [19]. The back-end returns the visual data needed to the front end for rendering the output visualization.

### 6.1 Blocks interface

The Blocks interface provides a visual, drag-and-drop technique to encode fields, consistent with **DG1**. Like Tableau, pills represent fields and a schema pane contains the list of fields from the connected data source. Instead of an interface of a fixed number of shelves, the Blocks interface provides a canvas that supports an arbitrary number of Blocks. Dragging out a pill to a blank spot on the canvas will create a new Block, defaulting the Block's encoding, mark type and layout type based on metadata of the field that the pill represents.

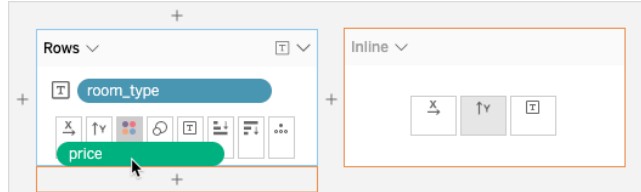

Figure 3: Possible drop targets are shown to the user just-in-time as they drag pills to the Blocks canvas.

For example, dragging out a pill that represents the discrete, string field P will create a Block with the layout type of rows, mark type of text, and field P encoded on ⊤ . The layout type and mark type are displayed at the top of the Block. Encodings are displayed as a list inside the block. Additional pills can be dragged to blank space on the canvas to create a new, unrelated block, added as an additional encoding to Block A, or dropped adjacent to Block A to create a new related block.

As seen in Figure 3, when a pill is dragged over an existing block, drop targets appear that represent any unused encodings in that Block that the system provides. When a pill is dragged over an area adjacent to an existing block, drop targets appear to assist in creating a new related block. If the pill that is being dragged represents a dimension field, the system provides options to create a new block with either the rows layout type or the column layout type. The dimension field of the pill will be encoded on ⊤ by default. If the pill being dragged represents a measure field, the system provides the option to encode the measure on the ⤴ , ↑Y , or ⊤ on a Block that is defaulted to the inline layout type. Once the new, related Block is created, the layout type, mark type, and encoding can all be customized.

There are two implicitly-created root Blocks that are invisible in the interface, a Rows root and a Columns root. Any Block that has no parents is the child of the Rows root Block and Columns root Block. These root Blocks are used as the starting point for calculating Row and Column indexes, as described in Section 6.3.

Blocks placed to the right of or below related Blocks are automat-

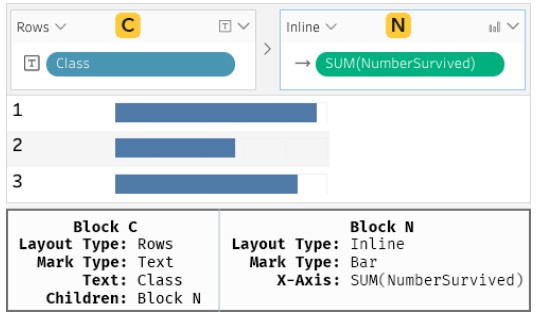

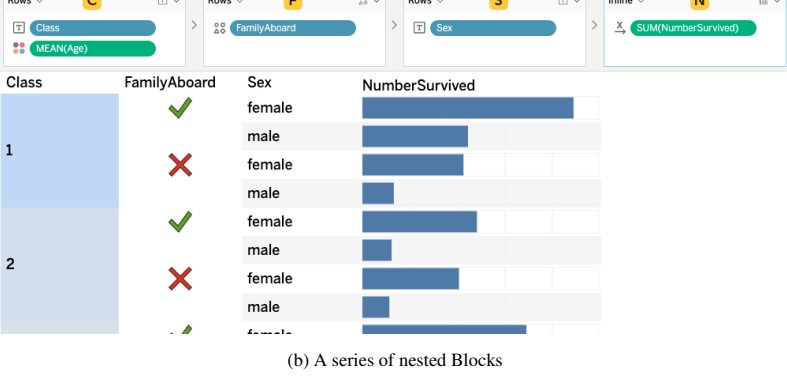

(a) The user interface representation and the formalism representation of two Blocks. Block C, a Rows Block encoding `Class`, facets Block N, an inline Block showing `SUM(NumberSurvived)` as a bar mark.

(b) A series of nested Blocks

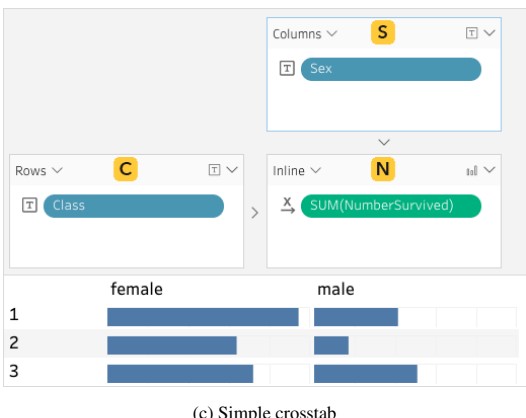

(c) Simple crosstab

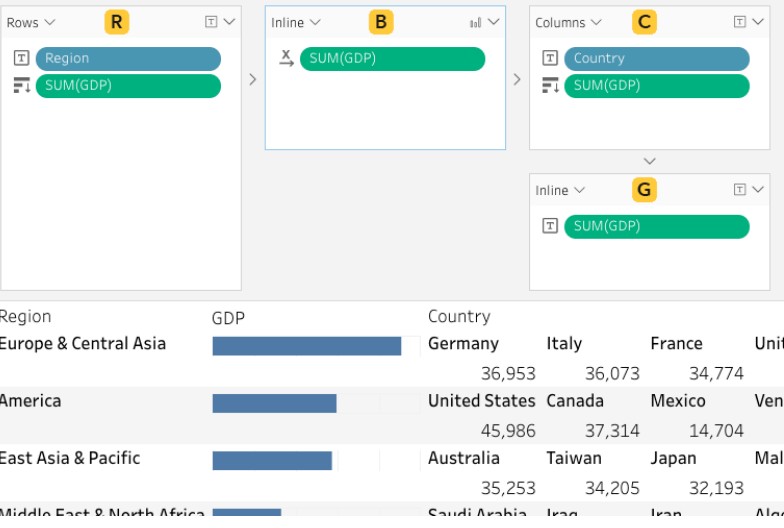

(d) Defining nested columns

Figure 4: Example configurations of Blocks

ically determined to be child Blocks. A chevron icon (>) displayed between the Blocks denotes the direction of the nested relationship between the Blocks. The layout of Blocks also directly determines the layout of components in the visualization. Block A placed above Block B will draw visualization component A above visualization component B.

Every Block must have both a Rows and a Columns parent to determine its position in the visualization; more than two parents are not permitted. If a Rows or Columns parent is not explicit in the interface, that parent is added implicitly by the system. The missing parent is implied by the relationships of the defined parent. A Block that does not have a Columns parent defined in the interface uses the Column parent of its Rows parent. Similarly, a Block that does not have a Rows parent defined in the interface uses the Rows parent of its Columns parent. Inline Blocks do not have children. If the interface defines a Block as a child of an Inline Block, it uses the Rows and Columns parents of the Inline Block. Figure 5 shows the graph implied by the interface for Figure 4d.

In Figure 4a, the field `Class` is encoded on text in Block C with a Row layout type. As there are three values in the domain of the field of `Class`, three rows are created in the visualization with a text mark for each value of the field. An Inline Block is nested as a child Block with `NumSurvived` encoded on the ⤳ . The system creates a bar chart for each row as defined by the first Block. Since no additional dimensions are added to Block N, the measure `NumSurvived` is aggregated to the LOD of `Class` and a single bar is rendered per

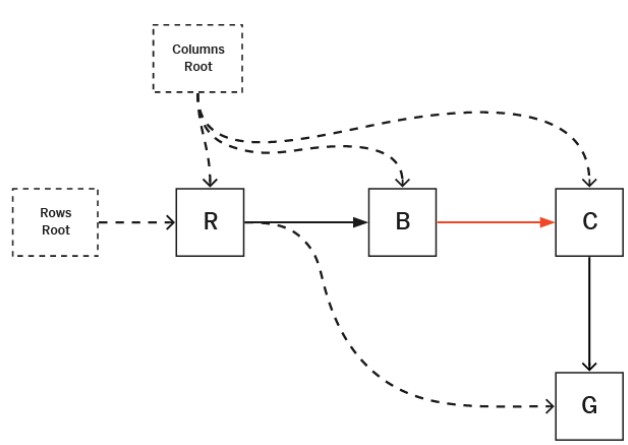

Figure 5: Implicit and explicit links for Figure 4d. Links explicitly shown in the interface are solid black arrows. The link from an Inline Block, which is treated as a link from the parent block, is shown in red. Links added implicitly are shown as dashed arrows.

row.

Figure 4b expands the example, showing how multiple dimensions can be added to a visualization. Due to the parent-child relationship of the four Blocks, `NumSurvived` inherits the dimensions from the parent Blocks, and aggregates at the combined LOD of `Class`, `Family Aboard`, and `Sex`. In contrast, `Age` is encoded in Block C which has no parent Block. Therefore, `Age` aggregates to the LOD of `Class`, the only dimension encoded on the same Block.

To specify a crosstab, the formalism requires a Block to have two parents - a Rows Block parent and a Columns Block parent. The user interface supports the specification of two parent Blocks, one directly to the left and the other directly above a Block. Figure 4c shows Block N with Block C and Block S as parent Blocks. The measure of `NumSurvived` in Block N is aggregated to the combined LOD of `Class` and `Sex`, the dimensions from its parents, Block C and Block S.

### 6.2 Query execution

Each Block executes a single query that is at the LOD of all the dimensions for the Block, including those inherited from parent Blocks. In figure 4b, the query for Block *S* includes not only `Sex` but also `FamilyAboard`, inherited direction from Block F and `Class`, inherited indirectly from Block C. This enables layout of the Block's marks relative to its parents and avoids making the user repeat themselves in the user interface, in support of **DG1**. The query includes only the measures for the current Block, not those of any other Block, because measures are aggregates at a specific LOD, in support of **DG3**. Every query is deterministically sorted, either by a user-requested sort or by a default sort based on the order of the encoding fields within the block.

### 6.3 Row and Column Assignment

Query execution results in multiple tables with different schemas. The system needs to assign Row and Column indexes from a single grid to tuples from all of these tables. This section describes the process for Rows; it is repeated for Columns.

1. Produce a Block tree from the Blocks DAG by only considering links from Rows Blocks to their children, excluding any other links. The Blocks Interface ensures that this tree exists, is connected, and has a single root at the implicit Rows root Block.

2. Produce a tuples tree by treating each tuple as a node. Its parent is the tuple from its parent Block with matching dimension values.

3. Sort the children of each tuple, first in the order their Blocks appear as children in the Blocks tree, and then in the order of the Rows dimensions and user-specified sorts, if any, for each Block.

4. Assign row indexes to each tuple by walking the tuple tree in depth-first order. Leaf tuples get a single row index; interior nodes record the minimum and maximum row indexes of all their leaves into the tuple.

### 6.4 Output visualization

Each tuple from a Rows or Columns Block forms a single cell containing a single mark. All of the tuples from an Inline Block with the same Row and Column parent tuples form a single cell. The values of visual encoding fields that are dimensions, if any, differentiate between marks within that cell. Those marks may comprise a bar chart, scatter plot, or other visualization depending on the mark type and visual encodings of the Block. The system uses a CSS Grid [9] and the computed row and column minimum and maximum indexes to define the position of each cell. Within each cell, simple text marks are rendered using HTML. A SVG-based renderer is used for all other marks.

## 7 COMPARATIVE STUDY OF BLOCKS WITH TABLEAU

We conducted a user study of Blocks with the goal of answering two research questions: **RQ1**: How do users orient and familiarize themselves with the Blocks paradigm? and **RQ2**: What are the differences in how users create visualizations across Tableau and Blocks? This information would provide insights as to how Blocks could be useful to users and how the paradigm could potentially integrate into a more comprehensive visual analysis system. The study had two parts: Part 1 was an exploratory warm-up exercise to observe how people would familiarize themselves with the Blocks interface in an open-ended way. Part 2 was a comparative study where participants completed an assigned visual analysis task of creating a visualization using both Tableau and Blocks. The study focused on various rich table creation tasks as they were found to be a prevalent type of visualization as described in Section 4. Comparing Blocks with Tableau would help highlight the differences in the participants' analytical workflows when performing the same task.

### 7.1 Method

#### 7.1.1 Participants

A total of 24 volunteer participants (6 female, 18 male) took part in the studies and none of them participated more than once. All participants were fluent in English and recruited from a visual analytics organization without any monetary incentives. The participants had a variety of job backgrounds - user researcher, sales consultant, engineering leader, data analyst, product manager, technical program manager and marketing manager. Based on self-reporting, eight were experienced users of the Tableau product, eight had moderate experience, while eight had limited proficiency. During Part 2 of the study, each participant was randomly assigned an order of whether to use Blocks or Tableau first when completing their assigned task.

#### 7.1.2 Procedure and Apparatus

Two of the authors supported each session, one being the facilitator and the other as the notetaker. All the study trials were done remotely over a shared screen video conference to conform with social distancing protocol due to COVID-19. All sessions took approximately 50 minutes and were recorded. We began the study with the facilitator reading from an instructions script, followed by sharing a short (under two minutes) tutorial video of the Blocks interface, explaining the possible interactions. Participants were then provided a URL link to the Blocks prototype where they participated in Part 1 of the study using the Superstore dataset [24]. During this part, they were instructed to think aloud, and to tell us whenever the system did something unexpected. Halfway through the study session, participants transitioned to Part 2 of the study. They were provided instructions to the task to perform with a Tableau Online [21] workbook pre-populated with the dataset and the Blocks prototype. We discussed reactions to system behavior throughout the session and then concluded with a semi-structured interview. Experimenter script, task instructions, and tutorial video are included in supplementary material.

#### 7.1.3 Tasks

There were two main parts to the study: Open-ended exploration and closed-ended tasks.

**Part 1: Open-ended exploration** This task enabled us to observe how people would explore and familiarize themselves with the Blocks interface. Instructions were: "Based on what you saw in the tutorial video, we would like you to explore this data in the Blocks prototype. As you work, please let us know what questions or hypotheses you're trying to answer as well as any insights you have while using the interface."

**Part 2: Closed-ended tasks**

The closed-ended tasks were intended to provide some consistent objectives for task comparison across both Tableau and Blocks systems.

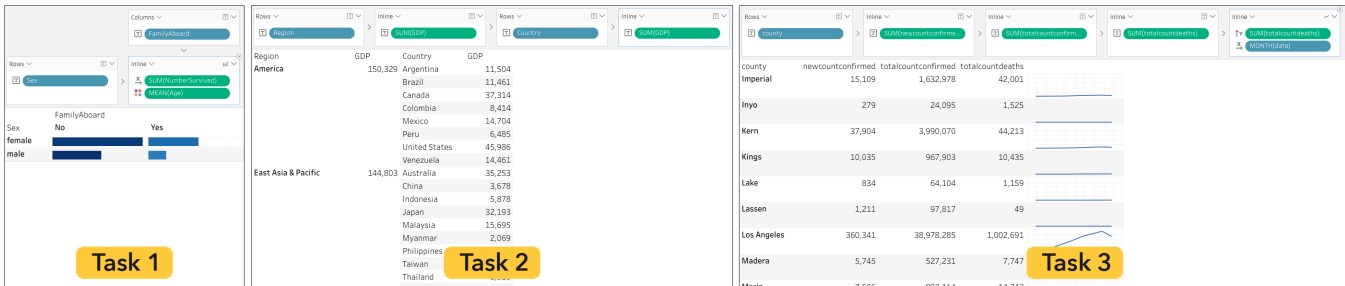

Figure 6: Three study tasks. Task 1: Cross tab with bar charts, Task 2: Table with sorted dimensions, and Task 3: Table with sparklines

Participants completed one of three randomly assigned closed-ended tasks that involved the creation of a rich table as shown in Figure 6. Expected visualization result images were shown as visual guidance along with the instructions to indicate what was generally expected as part of task completion. Here are the tasks along with their corresponding instructions that were provided to the participants:

- **Task 1: Create a crosstab with barcharts** "Using the Titanic dataset [10], create a crosstab for SUM(NumberSurvived) by Sex (on Rows) and FamilyAboard (on Columns). Now, switch to show bar charts for NumberSurvived with AVG(Age) on color."
- **Task 2: Create a sorted table** "Using the Gapminder dataset [30], create a table that shows SUM(GDP) for each Region and Country. Now, using the table from the previous step, sort both Region and Country by SUM(GDP)."
- **Task 3: Create a table with sparklines** "Using the COVID-19 dataset [8], create a table that shows New Count Confirmed, Total Count Confirmed, and Total Count Deaths for each County in California. Now, given the table from the previous task, add a column with the time attribute Date to generate sparklines to show the Total Count Deaths over time."

### 7.1.4 Analysis Approach

The primary focus of our work was a qualitative analysis of how Blocks influenced people's analytical workflows and comparing those workflows with that of Tableau. We conducted a thematic analysis through open-coding of session videos, focusing on strategies participants took. Given the remote nature of the study setup, we did not measure the time taken for task completion. We use the notation *PID* to refer to the study participants.

## 8 STUDY FINDINGS

### 8.1 RQ1: How do users orient and familiarize themselves with the Blocks paradigm?

To understand how intuitive the Blocks paradigm is for users, we first examine the strategies participants adopted for sense-making as they oriented themselves with the workings of the interface during Part 1 of the study. We observed various assumptions, expectations, and disconnections users faced as they drew from their past experiences while developing their own mental models when exploring Blocks.

### 8.1.1 Expectations with drag-and-drop interaction

When asked to explore the Blocks interface, all participants immediately dragged attribute pills from the data pane onto the canvas; a paradigm that many of them were familiar with having used Tableau and PowerBI. P4 remarked while using the Superstore dataset - "I'm going to drag Category on the canvas and let it go and I see that it created a Block showing the various category values." When subsequent attributes were dragged on to the canvas, several participants

were initially uncertain what the various drop targets were and how dropping a pill onto a new Block would affect the other Blocks currently on the canvas (P15). P4 - "I'm dragging out a new pill and I see these various drop targets, but do not know the difference between these." They eventually discovered that there are multiple drop targets within each Block for the various encodings as well as drop targets above, below, to the left and right of each Block. Participants often dragged Blocks around the canvas to change the structure of the generated visualization. Some participants (P4, P5, P6, P9, P11) wanted to modify the current Blocks on the canvas by dragging pills from one Block to another. When they realized that the interface currently does not support that functionality, they deleted the pill in one Block and dragged out the same pill onto another Block to replicate their intention.

### 8.1.2 Understanding the concept of a Block

While the Blocks interface has some commonalities with that of Tableau's interface around marks and encodings, there are differences that participants took some time to understand. In particular, the Blocks interface moves away from the shelves paradigm in Tableau. It relies on users to set encoding properties within each Block for each mark type, and the layout is defined by the relative positions of other Blocks on the canvas. P4 tried to externalize her mental model of the interface, reconciling against that of Tableau, "I'm just trying to wrap my head around this. Looks like we are not constrained here [Blocks] by the rows, columns, marks paradigm from Tableau. I created rows with Category and I kept trying to drop Sales on Rows too, and now I notice these little arrows to drop on x or y."

Participants were initially unclear what the effects of the x- and y-axes encodings were on data values within a Block. P2 for example, set the mark of the Block to 'bar' and expected SUM(Sales), which was set on text encoding, to be displayed as a bar chart. After being guided by the experimenter to change the encoding from 🔲 encoding to ⤢ encoding, the semantics of the encoding properties became clearer. Other participants thought that the way to set encoding properties in the Block had a direct relationship to what they saw in the corresponding chart that was generated. P9 said, "This [Blocks] is much more literal. If I want to affect the Profit bar, I need to literally put the color on the Profit bar. In Tableau, I think of coloring the Category by Profit."

### 8.1.3 Direct manipulation behavior

The visual drop targets around a Block in the interface piqued participants' curiosity in exploring what would happen when they dragged out pills to these targets. P8 remarked, "I'd like to get an intuitive sense as to what happens when I drop it here [pointing below the Block] or there [pointing to the right]." Participants were able to understand the relationship between adding Blocks horizontally and the effects on the generated chart. Placing Blocks below one another took further exploration to better understand the system behavior. P11 said, "Going across seems straightforward. I'm trying to figure out what going down meant" and followed up his inquiry by adding

Blocks below an existing Block using the various layout options. P19 adopted a strategy of updating a Block with all the desired encodings – "I'm building out one definition for the first column of rows and then do the next." Participants also found it useful to be able to modify the existing chart by dragging pills into new Blocks in the middle of or adjacent to other Blocks, breaking down attributes into targeted levels of detail immediately. They found the visual layout of the Blocks to directly inform the structure of the generated chart – "The LOD of what is to the right is defined by what is to the left [P2]" and "You build out the viz literally the way you think about it [P6]." For some participants, the system did not match their expectations of how a dimension would be broken down by a measure. P8 said, "I put SUM(Sales) below Category and I expected Category to be broken down by Sales, but it showed me a single aggregated bar instead."

## 8.2 RQ2: What are the differences in how users create visualizations across Tableau and Blocks?

### 8.2.1 Task 1: Create a crosstab with barcharts

All eight participants were able to complete the task in both Blocks and Tableau. Here, we describe the workflows for both Blocks and Tableau.

*Blocks:* Adding text values for NumberSurvived in the table was relatively easy for all the participants. Participants took some time to figure out how to get the headers to appear in the expected spots (P2, P6). Putting Sex to the left of the current Block helped orient the participants with Block placement to generate the headers. All participants found it straightforward to then add bar charts by changing the encoding of NumberSurvived to ⤭ and adding AVG(Age) on ⠿ in the Block. P9 realized that the placement of Blocks is a literal translation to the placement of headers in the visualization and was able to add the headers looking at the visual provided as a reference.

*Tableau:* For participants fluent with using Tableau, creating the crosstab was a quick task. Participants first built the rows and columns in the crosstab and then added a measure. This workflow conflicted with the way participants (P12) created a crosstab in Block, where they started with adding the measure first. P2 said, "In Tableau, the fact that the headers are inside Columns and Rows than being in some separate place like in Blocks, makes it easier to generate." P9 struggled a bit to add barcharts to the crosstab and mentioned that it is was not very intuitive to place SUM(NumberSurvived) on columns.

### 8.2.2 Task 2: Create a sorted table

All eight participants were able to complete the task in Blocks. Two participants (P8 and P14) needed guidance to complete the task in Tableau. Here, we describe the workflows for both Blocks and Tableau.

*Blocks:* All the participants dragged out the pills in the order of the columns in the table – Region, Country, and GDP with the encoding set to ⊞ . They were able to complete the task quickly and appreciated the fact that they did not have to write a calculated field and the LOD was computed automatically based on the relative positions of the Blocks. P11 said, "That's cool. The LOD did what I would've expected if I wasn't used to using Tableau." P3 commented, "It seems like we need new Blocks for each partition aggregation." It was not immediately intuitive for a few participants as to how Region and Country needed to be sorted by GDP. Eventually when they dragged the GDP pill to the Region and Country Blocks, they noticed a sort icon appear and realized that sorting of a dimension is performed per Block.

*Tableau:* A prevalent technique that participants employed was using a calculated field (P3, P5, P11, P17, P20). Participants first added the Region and Country dimensions to Rows with the GDP measure added as text. They then created a calculated field for GDP per Region at the level of Region and converted it into a discrete pill

in order to add it between two dimensions, Region and Country in the table. All participants took advantage of Tableau's contextual menu by right-clicking on the table's headers to sort the values in descending order.

### 8.2.3 Task 3: Create a table with sparklines

All eight participants were able to complete the task in Blocks. One participant (P7) was unable to add sparklines to the table in Tableau. Here, we describe the workflows for both Blocks and Tableau.

*Blocks:* All participants dragged out County, New Count Confirmed, Total Count Confirmed, and Total Count Deaths into separate Blocks that were laid out horizontally. Generating a column of sparklines in the table was easy for all participants; they intuitively dragged Date onto the ⤭ encoding and Total Count Deaths onto the ↑γ encoding into a new Block.

*Tableau:* All participants created the initial table with Tableau's Measure Names[1] and Measure Values[2] fields using County as the dimension to group the data by. Adding a column containing sparklines was more challenging for all participants. P4, P10, P16, and P22 created LOD calculations for each of the three measures New Count Confirmed, Total Count Confirmed, and Total Count Deaths, making each calculated field discrete so that the values could be broken down by County. Line charts were added to the table using Total Count Deaths over Date. P13 and P19 were unsure how to add sparklines to the existing table; they used a different approach by creating a separate worksheet containing a column of line charts and placed it adjacent to the initial table in a Tableau dashboard.

## 8.3 Discussion

General feedback from the participants was positive and suggested that Blocks is a promising paradigm to have more control over the layout and manipulating the LOD in the structure of the created visualization. Participants identified certain tasks that could take longer to do in a tool like Tableau, that would be easier in Blocks. P12 remarked, "This is ridiculously awesome. I'm not going to lie, but I have this horrific cross tab bookmarked to do in Tableau. I can see doing it in Blocks in a minute and a half." Participants appreciated the flexibility of being able to apply conditional formatting to various parts of a visualization and not just for the measures. P19 commented, "That's cool. I've never been able to do conditional color dimensions before." Having more control over LOD was a consistent feature that participants found useful. P6 said, "You can do all these subdivisions that are hard to do in Tableau." and "Aha! I can get sparklines so easily." P2 said, "The fact that I can put all these encodings in Blocks makes it a heck of a lot more expressive." Participants also used the canvas to create different visualizations by laying out arrangements of Blocks in space, akin to a computational notebook. The layout helped them compare arrangements with one another as they reasoned about the effects of visual arrangement on chart structure. P15 commented, "In Tableau, I am forced to create a single visualization in each worksheet and then need to assemble them together into a dashboard. In Blocks, it feels like a canvas where I can create how many ever things I want."

There were some limitations that the participants brought up with the Blocks prototype.

### 8.3.1 Need for better defaults and previews

The flexibility that the Blocks interface affords also comes with an inherent downside of a vast set of drop-target options. P10 was overwhelmed with the choices when he initially started exploring and remarked, "There are so many arrows to choose from. It would be helpful if I can get a hint as to where I should drop by pill

---

[1]The Measure Names field contains the names of all measures in the data, collected into a single field with discrete values.

[2]The Measure Values field contains all the measures in the data, collected into a single field with continuous values.

based on what attribute I selected." Others wanted to see chart recommendations based on the pills they were interested in, similar to Show Me [3] in Tableau – "Would be nice to get a simple chart like Show Me by clicking on the attributes [P4]." P6 commented, "It would be nice if Blocks could just do the right things when I drop pills onto the [Blocks] canvas."

Showing previews and feedback in the interface when users drag pills to various encoding options within a Block or when new Blocks are created, could better orient the user to the workings of the interface. P12 suggested, "It would be really cool if there are actions associated with the visual indicators of the drop targets so the interface does not feel too free form." For example, dragging `Age` to a Block could highlight the particular column or cell in the visualization that would be affected by that change. P5 added, "I tend to experiment around and having previews show up as I drag pills to drop targets, would be helpful." Providing reasonable defaults such as suggesting a ↑y encoding for a pill when the Block already has a ⤩ encoding, could help guide the user towards useful encoding choices.

### 8.3.2 More control over chart customization

Participants wanted additional customization in the interface. P3 said, "It would be nice if I could center the sparklines to the text in the table. I would also like to add a dot on the maximum values in the sparklines." Showing hierarchical data in a table requires Blocks to be added for each level that can potentially take up significant screen real-estate for large hierarchies. One workaround suggested was incorporating a Tableau UI feature to drill down into a hierarchical field within a Block (P13). The Blocks prototype also currently lacks templating actions such as adding borders and formatting text in headers that participants were accustomed to in Tableau (P12).

### 8.3.3 Support for additional analytical capabilities

Participants wanted more advanced analytical capabilities such as calculated fields to add additional computations to the visual panes in the charts. P3 remarked, "I'd like to use a table calculation[4] to add a max sales values or running totals for that block." Others wanted the prototype to support additional chart types such as maps (P19, P20).

## 9 LONGITUDINAL DIARY STUDIES

One of the limitations of the comparative studies was that participants had more experience with using Tableau than with Blocks. Our previous study focused on how Blocks were used in the short term during a single lab session. We offered an option to our study participants to take part in a two-week diary study. The goal of the diary study was to better understand users' behavioral patterns over a longer period of time and how they would use Blocks in their own exploratory analyses. In total, eight participants (seven male and one female) took part in the study where they documented their experiences using the Blocks prototype in Google Docs, spending at least 20 minutes a day for two weeks. Similar to the analysis approach in the previous user study, we conducted a thematic analysis through open-coding of the diary notes. The actual diaries are included as part of supplementary material.

### 9.1 Diary Study Observations

Participants appreciated the ease of use of creating more complex rich tables. P3 found that this task was easier to do in Blocks than in Tableau – "Now I want to add more measures in this small multiples, which is super hard when you want to do this with >2 measures in

---

[3] Show Me creates a view based on the fields in the view and any fields you've selected in the data pane.

[4] A type of calculated field in Tableau that computes based on what is currently in the visualization and does not consider any measures or dimensions that are filtered out of the visualization.

Tableau. With Blocks I can easily add as many as I want within the partition I'm interested in." P20 commented, "There is something to be said for how easy this type of thing is. Multi sparklines alongside totals shown in multiple perspectives." The extended period of time to explore the prototype also helped participants to reflect upon their understanding of how Blocks worked. P9 summarized by saying, "It seems like the mental model in Blocks is 'Which number are you interested in?' You start with that, then you start breaking it down dimensionally to the left/right/top/bottom. In Tableau, I go to the dimensions first and then drop in my measure later. Both of these make sense, but I would like to get to a point where I can use my old mental model (dimensions first, then measures) and still be successful in Blocks. Sometimes I know my dimensionality first – voting by age/gender/precinct – I want to drop that in and then look at the measures."

There were also aspects of the prototype that were limiting to participants' exploratory analyses. Suggesting smart defaults in the Blocks interface continued to be a theme in the participants' feedback. P1 documented, "It would be helpful if Blocks can guide me towards building useful views. For example, I'm using the Superstore data source, and when I drag out `Category` and `Profit`, it would be useful to suggest the x-axis, showing horizontal bar charts that combine the headers and the bars nicely." P3 had a suggestion about better encoding defaults – "I first dropped a measure to create a block, I got a text mark type by default. But it would have been nice to pick up Circle or something similar to make the size encoding meaningful".

Some participants wanted interaction behaviors from Tableau in the prototype such as double-clicking to get a default chart similar to Show Me. P2 said, "I wanted to double-click to start adding fields instead of drag and drop. Especially for the first field when I'm just exploring the data. I'd also like to able to scroll the chart area independently of the Blocks". Participants (P2, P18, P20) tried to create other chart types such as stacked bar charts, tree maps, and Sankey charts that Blocks did not support at the time of the study.

## 10 BEYOND TABLES: OTHER USE CASES & FUTURE WORK

In this paper, we demonstrate how the Blocks formalism can be used to create complex rich tables. Blocks can be extended to support other visualizations such as treemaps, bubble charts and non-rectangular charts with additional layout algorithms. Blocks does not currently support layering or juxtaposed views that are prevalent in composite visualizations. Future work could explore how to support the creation of these visualizations in the Blocks interface. The ability to define rich tables at multiple LODs could be applied to support other visualization types such as Sankey diagrams and composite maps.

Sankey diagrams are a common type of chart created in Tableau, but the creation is a multi-step process involving partitioning the view, densifying the data, indexing the values across different visual dimensions, and several table calculations [6]. With the Blocks system, an $n$-level Sankey diagram could be built with $2n$-1 Blocks as shown in Figure 7: the Row Blocks represent the nodes of the Sankey for `Region`, `Category`, and `Segment` attributes, while the Link Blocks represent connecting between levels. The Link Blocks inherit their LOD from the neighboring Blocks and render the curves between pairs of marks. The links are encoded by color and size based on `SUM(Sales)`.

The composite map visualization in Figure 8 shows `State` polygons as parent Blocks and nested sparkline charts containing `Sales` by `Order Date`. The visualization is constructed using an Inline Block for the map with the sparkline Block as its child.

While Blocks employs direct manipulation for supporting the creation of expressive charts, there is an opportunity to add scaffolds through thoughtful defaults and previews to better support users and their mental models when learning the workings of the new

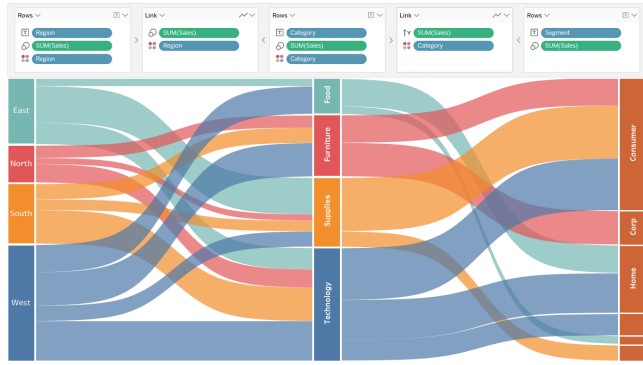

Figure 7: A two-level Sankey Diagram

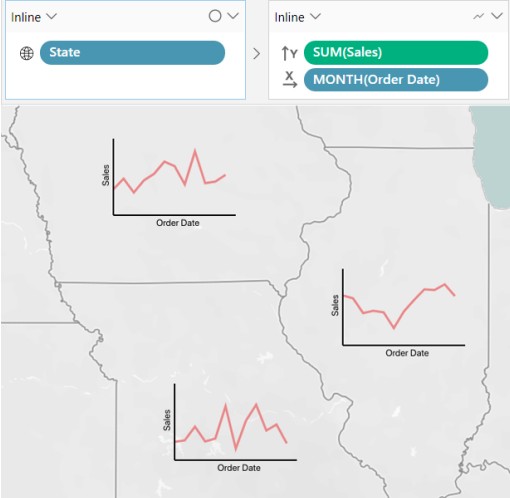

Figure 8: Map with nested sparkline charts

interface. We would like to explore how visual interaction during chart generation can be better supported by bridging the user's intentions with the facilities afforded by the interface. The Blocks interface shows promise in supporting analytical workflows that are currently challenging to perform in Tableau, but additional analytical capabilities such as new chart types, support for reference lines, and better formatting options need to be incorporated to be truly useful. Exploring the balance between comprehensive analytical capabilities yet reducing friction in accomplishing users' goals, is an important research direction to pursue.

We evaluated Blocks with users who had varied degrees of familiarity using Tableau. The study findings indicate that their mental models when exploring the Blocks interface were influenced in part by their prior experience with the Tableau interface. While Blocks and Tableau share some common paradigms, they do have differences. As we continue to evolve Blocks, we would like to further evaluate how the effects of reality and expectations cross with users who have no experience using Tableau compared to their counterparts who frequently use Tableau. Understanding how users create new mental models or upgrade existing ones would help inform ways to support effective onboarding to the Blocks paradigm.

## 11 CONCLUSION

We present Blocks, a new formalism that builds upon VizQL by supporting the handling of nesting relationships between attributes through direct manipulation. By treating each component of the visualization as an analytical entity, users can set different LOD

and encoding properties through drag-and-drop interactions in the Blocks interface. An evaluation of the Blocks interface and comparing users' analytical workflows with Tableau indicates that Blocks is a useful paradigm for supporting the creation of rich tables with embedded charts. We further demonstrate how Blocks is generalizable to express more complex nested visualizations. Future research directions will explore additional analytical and interaction capabilities in the system along with useful scaffolds for supporting users during visual analysis. We hope that insights learned from our work can identify interesting research directions to help strike a balance between expressivity, ease of use, and analytical richness in visual analysis tools.

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
