# OpenReview forum: "Blocks: Creating Rich Tables with Drag-and-Drop Interaction"
_graphicsinterface.org/Graphics_Interface/2022/Conference — Submitted to GI 2022_

### Official Review · Reviewer_DRdw · 2022-01-08
**This paper presents a demonstration of a system for creating drag-and-drop visualizations, but the user study is rather superficial.**

**Rating:** 4
**Confidence:** 4

**Review:**

This paper presents a drag-and-drop approach for generating visualizations, and is compared to Tableau for performing similar visualization generation. The first half of the paper presents a reasonably good demonstration of the system. Unfortunately, the user studies are rather superficial.

The primary contribution of this work is the introduction of direct manipulation (drag & drop) into the visualization creation process. However, Tableau also supports such interaction, yet within tighter constraints for where things can be dropped. One of the things that is missing from this work is an analysis of the academic literature with respect to such direct manipulation, and a discussion of the benefits and drawbacks of such an approach. Further analysis of how different visualization frameworks such as Tableau support such direct interaction would be beneficial, so that the specifics of what is different in Blocks can be explained.

There are many statements made throughout the first few pages that are asserted without proper reference (e.g., "that add both complexity and friction to the analytical workflow"). Looking to the reference section, there are many technical references and few academic references, making this appear to be more of a technical report than an academic paper.

There are a number of important points made in the Related Work section that are motivation for the work done in the design and implementation of Blocks (e.g., "focus on supporting non-developers" and "handling nested relationships during chart generation"). Further, there are some comparisons made to the work in this paper (e.g., last paragraph of Section 2.2) that are hard to make sense of since the reader has not yet seen what the proposed approach is. I suggest keeping the Related Work section focused on what others have done, and moving the motivation to the introduction and the comparison to other works to the discussions.

While it is great to see that interviews, artifact analysis, and discussion board analyses were conducted to collect design goals. However, references for the qualitative methods used, and methodological information such as the interview guides are missing. For the first design goal, there seems to be a disconnect between the discussion about table-based visualizations and the drag-and-drop interaction. The qualitative analysis of this information is not well explained or supported.

The presentation of the Block system itself is well done. My only criticism of this part of the paper is that the authors refer to Figure 4 before Figure 2 is introduced. Up to here, I think this is could be a reasonably good demonstration paper, if the introduction and background sections are enhanced with more academic literature to justify the assertions and design decisions.

Unfortunately, the two studies appear to be rather superficial. The study designs need to be explained more precisely so that the reader can determine how rigourously the study was conducted. For example, issues of within-subjects/between-subjects designs, independent variables, and dependent variables should be explained. Instructions for the think-aloud protocols and the diary study are needed. The interview questions for the semi-structured interview should be provided.

A major concern I have is the disparity of prior Tableau experience among the participants. Since Tableau is the baseline against which Blocks is compared, the difference in prior experience represents a confounding factor. This can easily be dealt with by considering this a quasi-independent variable and analyzing the results of each group of participants separately. Unfortunately, this was not done. As such, the reader does not know if a particular finding is due to a participant's high level of Tableau experience, because it is low, or if it is unrelated.

It seems as though some of the tasks might be rather difficult to perform using Tableau (especially for the novice Tableau users). Perhaps there needs to be some discussion about the difficulty and complexity of undertaking these tasks with the two interfaces (e.g., in a perfect situation, how many steps would it take a user to perform the assigned tasks?).

For the qualitative analysis, it is stated that open coding/thematic analysis is conducted, with a focus on the participants' strategies. However, the findings are generally just descriptions of the steps/process generally taken to complete the tasks, with very little discussion of the strategies used. There are no codes or themes explicitly stated in the findings. While the explanation is peppered with participant quotes, it is unclear if these came from things the participants said while performing the tasks, or the interviews. These same issues continue into the diary study.

While it is clear that Blocks represents an interesting alternative to using Tableau for certain types of visualization creation, the comparison to Tableau via the qualitative user study was not conducted in a sufficiently rigorous manner to have the findings be trustworthy. I think there is an opportunity for the authors to conduct a user study in a more careful, methodical, and rigorous manner to show the benefits and drawbacks of one system over the other.

---

### Official Review · Reviewer_huWN · 2022-01-10
**Interesting read, but worry about generalizability**

**Rating:** 6
**Confidence:** 4

**Review:**

This paper presents the Blocks formalism, a specification that enables users to declare and manipulate the layouts, data relationships, and level of details that are displayed within different segments of a visualization that is created in Tableau. The formalism itself is described, along with design goals that were derived from an interview-based study with Tableau customers. Then, the paper describes the results of a usability study comparing Tableau with Tableau+Blocks, and also the results from a longitudinal diary study. The results seem to suggest that the Blocks formalism is a welcome addition to the Tableau software and solves some of its existing problems.

Note that I am not a data visualization expert, so please read the following review with this in mind.

Overall, I thought this paper was quite easy to read and the related work set up the contribution of the research well. The figures were also fairly easy to understand. I think the contribution fits the length of the paper and the overall goal of increasing the usability of tables with additional drag and drop functionality seems like an important and worthwhile goal. I do, however, have three concerns.

First, the paper doesn’t really build the case for why there needs to be additional support for nested and inheritance capabilities via drag and drop. The intro makes the case for utilizing drag and drop, but the motivation for using drag and drop for this specific aspect of visual analyses is missing.

Second, it is unclear why Tableau was chosen as the system to study and extend. Reading between the lines, it may be because the author(s) are part of the Tableau company (e.g., mentions of customers, dashboards, etc.), however, the paper needs to include more justification for why Tableau was used. From some perspectives, Blocks sort of comes across as an extension to Tableau that seeks to solve some of the software’s problems, but then this begs the questions, “What is the research question here”, “Is this simply an exercise in software engineering”, and “Can these findings be generalized to other data visualization software / tools?”. It is this second concern that is worrying to me because all of the findings are Tableau-centric and the text even says, “The study findings indicate that their mental models when exploring the Blocks interface were influenced in part by their prior experience with the Tableau interface.”

Lastly, Section 10 seems out of place and led me to wonder, “if Blocks supports the creation of such charts, why wasn’t this mentioned in Section 6 and why weren’t they evaluated in the studies?”. Rather than talking about additional functionality that is possible, perhaps the paper would be better served by the inclusion of a higher level synthesis of the findings from the last two studies outside the Tableau context – or at least a discussion of those findings that would not be generalizable.

Overall, this paper does have some interesting aspects, however, I do worry about its reliance and attachment to the Tableau software. For this reason, I am on the fence about acceptance.
Smaller Concerns:
-	The last paragraph in the intro seems out of place
-	How were all study participants recruited, were they paid, what were their ages?
-	Figure 2 should probably have 1-4 numbers somewhere in the caption
-	Part 1: Open-ended exploration has some formatting / font issues
-	Italicizing participant quotes makes them easier to read

---

### Official Review · Reviewer_qZFw · 2022-01-14
**Nice extension to visualization formalisms, but a bit thin on details**

**Rating:** 8
**Confidence:** 4

**Review:**

The paper describes Blocks, an extension to the VizQL formalism used in Tableau. The paper's contribution is the description of the Blocks idea and two evaluations showing that the extension is understandable and potentially valuable compared to the standard Tableau system.

Overall I like the idea and the paper. The Blocks idea is a good addition to the existing formalism, and is packaged well in a drag-and-drop interaction style. The paper is well written and easy to follow (although more figures are needed as mentioned below). The evaluations are appropriate for the stage of development of the tool (although with some limitations as mentioned below). The drawbacks are not so great that they compromise the paper's contribution, although there are ways that the authors could improve the paper prior to publication. These are:

1. Tableau-centric approach. I understand that the authors are focused on this ecosystem, but the research problem here is entirely motivated by the limitations of this particular formalism. The paper would be improved if the authors discussed how the ideas behind Blocks could be generalized to other formalisms (explicit ones, or implicit such as what is found in D3).

2. More illustrations needed. For readers who are not highly familiar with the Tableau interface and approach, the text descriptions are not enough to clearly indicate how the current formalism works and what the limitations are. In addition, the figures illustrating Blocks are not enough to clearly show what is going on (what the interactions are, what the overall system looks like, how the user does different tasks). More images would greatly assist the reader.

3. Comparative evaluation. The results of the comparison between Blocks and Tableau is too brief and too descriptive rather than indicating insights about the differences between the tools. In particular, the authors do not focus on why Blocks worked well (when it did) and what of these attributes are generalizable outside the specifics of Tableau. The authors need to clarify that it was some intrinsic aspect of Blocks that led to the successes (e.g., extending the formalism, drag-and-drop interaction, changing the basic components available to the user, etc.) rather than just e.g., a usability problem in Tableau. Second, the authors do not provide any information about the performance of the two tools, and this omission should either be remedied or justified. There are user comments suggesting that the Blocks system is faster than Tableau, and since the study seems to be set up in a comparative fashion, the lack of any performance data is an obvious omission.

Overall, the paper provides a lot of information, but to some degree it seems to be stretched too thin. The authors should at the very least use all of the space available to them to add detail, explanations for results, discussion of generalizability, and additional images.

---

### Decision · Program_Chairs · 2022-01-18

Reject